# Association of Germline Variation in Driver Genes with Breast Cancer Risk in Chilean Population

**DOI:** 10.3390/ijms242216076

**Published:** 2023-11-08

**Authors:** Sebastián Morales-Pison, Julio C. Tapia, Sarai Morales-González, Edio Maldonado, Mónica Acuña, Gloria M. Calaf, Lilian Jara

**Affiliations:** 1Centro de Oncología de Precisión (COP), Facultad de Medicina y Ciencias de la Salud, Universidad Mayor, Las Condes, Santiago 7560908, Chile; sebastian.morales@umayor.cl; 2Laboratorio de Transformación Celular, Programa de Biología Celular y Molecular, Instituto de Ciencias Biomédicas (ICBM), Facultad de Medicina, Universidad de Chile, Independencia, Santiago 783090, Chile; jtapiapineda@uchile.cl; 3Laboratorio de Genética Humana, Programa de Genética Humana, Instituto de Ciencias Biomédicas (ICBM), Facultad de Medicina, Universidad de Chile, Independencia, Santiago 783090, Chile; saraibet.morales@gmail.com (S.M.-G.); macuna@med.uchile.cl (M.A.); 4Programa de Biología Celular y Molecular, Instituto de Ciencias Biomédicas (ICBM), Facultad de Medicina, Universidad de Chile, Independencia, Santiago 783090, Chile; emaldona@med.uchile.cl; 5Instituto de Alta Investigación, Universidad de Tarapacá, Arica 1010069, Chile; gmcalaf@gmail.com

**Keywords:** breast cancer, driver gene, single nucleotide polymorphism, association study

## Abstract

Cancer is a genomic disease, with driver mutations contributing to tumorigenesis. These potentially heritable variants influence risk and underlie familial breast cancer (BC). This study evaluated associations between BC risk and 13 SNPs in driver genes *MAP3K1*, *SF3B1*, *SMAD4*, *ARID2*, *ATR*, *KMT2C*, *MAP3K13*, *NCOR1*, and *TBX3*, in *BRCA1/2*-negative Chilean families. SNPs were genotyped using TaqMan Assay in 492 cases and 1285 controls. There were no associations between rs75704921:C>T (*ARID2*); rs2229032:A>C (*ATR*); rs3735156:C>G (*KMT2C*); rs2276738:G>C, rs2293906:C>T, rs4075943T:>A, rs13091808:C>T (*MAP3K13*); rs178831:G>A (*NCOR1*); or rs3759173:C>A (*TBX3*) and risk. The *MAP3K1* rs832583 A allele (C/A+A/A) showed a protective effect in families with moderate BC history (OR = 0.7 [95% CI 0.5–0.9] *p* = 0.01). *SF3B1* rs16865677-T (G/T+T/T) increased risk in sporadic early-onset BC (OR = 1.4 [95% CI 1.0–2.0] *p* = 0.01). *SMAD4* rs3819122-C (A/C+C/C) increased risk in cases with moderate family history (OR = 2.0 [95% CI 1.3–2.9] *p* ≤ 0.0001) and sporadic cases diagnosed ≤50 years (OR = 1.6 [95% CI 1.1–2.2] *p* = 0.006). *SMAD4* rs12456284:A>G increased BC risk in G-allele carriers (A/G + G/G) in cases with ≥2 BC/OC cases and early-onset cases (OR = 1.2 [95% CI 1.0–1.6] *p* = 0.04 and OR = 1.4 [95% CI 1.0–1.9] *p* = 0.03, respectively). Our study suggests that specific germline variants in driver genes *MAP3K1*, *SF3B1*, and *SMAD4* contribute to BC risk in Chilean population.

## 1. Introduction

Breast cancer (BC) is the most prevalent cancer in women worldwide and the fifth leading cause of cancer deaths [1]. Globally, GLOBOCAN 2020 reported an estimated 2.26 million new cases and 685,000 BC deaths due to BC [1]. In women, BC accounts for approximately a quarter of all cancer cases and a sixth of all cancer deaths, ranking first for incidence in 159 of 185 countries and for mortality in 110 countries [2]. The incidence of female breast cancer is predicted to continue to increase globally, from 45.26 in 2010 to 47.36 in 2035 [2]. BC is the leading cause of death among women in Chile, with 2018 incidence and mortality rates of 46.2/100,000 and 16.6/100,000 women, respectively [3]. Recognized BC risk factors include gender, age, hormonal factors, and most significantly, family history. Genetic predisposition accounts for an estimated 5–10% of BC cases overall [4,5] and as much as 25% of cases in women under the age of 30 [6]. Although genetics play an important role in BC etiology, the major susceptibility genes *BRCA1* and *BRCA2* only explain about 16% of risk, and analysis of *BRCA1/BRCA2* mutations currently plays a highly significant role in oncological clinical genetics worldwide, with the goal of improving prevention and treatment for women at high risk. In Chile, studies have been carried out in relation to prevalent mutations in these genes, and this information is used in clinical practice [7,8,9]. More frequent but less-penetrant mutations have been identified in families with BC, located in moderate- or low-penetrance genes such as *CHEK2*, *RAD51C*, *RAD51D*, *ATM*, *PALB2*, *BRIP1*, *BARD1*, *TP53*, *CDH1*, *RECQL*, and *NBN* [10,11]. Recent research has focused on the low-penetrance variants that increase BC risk significantly in combination.

Cancer is a group of diseases characterized by abnormal and uncontrolled cellular growth, caused primarily by genetic mutations [12,13]. While numerous somatic mutations accumulate during tumorigenesis, the majority of these variants are neutral “passenger mutations”. Others, called “driver mutations”, confer somatic tissue cells with selective advantages over neighboring cells [12]. Such mutations occur in “driver genes”, the mutant forms of which affect the homeostatic development of a set of key cellular functions. The driver mutations and mutational processes operative in BC have not yet been comprehensively defined [14]. Since the establishment of the field of genetics, one of the main goals of cancer research has been to identify driver genes across tumor types [15,16,17,18]. Resulting discoveries have been applied to the development of targeted anticancer therapy paradigms and the search for genomic biomarkers of prognosis and treatment response [19]. Understanding the precise mode of alteration of each driver gene (namely, elucidating which of its mutations has the potential to drive tumorigenesis and why), as well as identifying the specific biological function perturbed in tumorigenesis, represent major forthcoming challenges in cancer genomics [20].

Several studies have used next-generation sequencing (NGS) to find potential driver mutations [14,21,22,23], including mutations specific to sporadic BC. Nevertheless, there has been scarce research into the possibility that these driver genes contain inherited variants that influence the development of cancer [24]. Göhler et al. [24] investigated whether known driver genes contain heritable variants that influence risk and/or survival in Swedish BC patients. The group evaluated selected single-nucleotide polymorphisms (SNPs) located in 15 genes consistently classified as BC driver genes by NGS. Five genes were associated with BC risk: *TBX3* (rs2242442) was associated with decreased risk; *TTN* (10497520) and *MAP3K1* (rs702688 and rs72758040) with increased risk; *MLL2* (rs11168827) with increased overall risk, positive hormone receptor status, and low-grade tumors; *SF3B1* (rs4688) with a protective effect as well as negative lymph node findings, metastasis, and hormone receptor status [25]. As the variations in these novel driver genes had not been assessed in any Latin American population, our group performed an association study on germline variations in BC driver genes in a Chilean sample. We evaluated associations between SNPs in the driver genes *TTN* (rs10497520), *TBX3* (rs2242442, rs8853, rs1061651, and rs12366395), *MLL2* (rs11168827), *MAP3K1* (rs702688, rs72758040, and rs702689), and *SF3B1* (rs4685) with BC risk in BRCA1/2-negative Chilean families [26,27].

The present study evaluates the association between 13 SNPs located in driver genes *MAP3K1*, *SF3B1*, *SMAD4*, *ARID2*, *ATR*, *KMT2C*, *MAP3K13*, *NCOR1*, and *TBX3* with familial and early-onset sporadic BC, studying cases and controls from Chilean families negative for BRCA1/2 point mutations. Using a case-control design, we analyzed associations between BC risk and the following SNPs: rs832583 (*MAP3K1*), rs16865677 (*SF3B1*), rs3819122 (*SMAD4*), rs12456284 (*SMAD4*), rs75704921 (*ARID2*), rs2229032 (*ATR*), rs3735156 (*KMT2C*), rs2276738 (*MAP3K13*), rs2293206 (*MAP3K13*), rs4075943 (*MAP3K13*), rs13091808 (*MAP3K13*), rs178831 (*NCOR1*), and rs3759173 (*TBX3*). Moreover, we assessed the SNP–SNP interaction between *SMAD4* rs3819122:A>C and *SF3B1* rs16865677:G>T to evaluate their combined effect on BC risk. The SNPs selected were chosen based on: (a) their presence in driver genes; (b) their locations within the genes; (c) their possible functional consequences. It is important to point out that there are scarce publications worldwide regarding the SNPs analyzed in the present manuscript.

## 2. Results

Table 1 shows the specific characteristics of the families of the 492 *BRCA1/2*-negative cases included in this study. Among the selected families, 63.4% had cases of both BC and ovarian cancer (OC). In the BC group, mean age at diagnosis was 42.8 years, and age of onset was ≤50 years in 36.6%. The observed genotype frequencies for 11 of the 13 polymorphisms were in Hardy–Weinberg equilibrium in controls (*p* = 0.25 for rs83258, *p* = 0.52 for rs2229032, *p* = 0.91 for rs3735156, *p* = 0.35 for rs2276738, *p* = 0.42 for rs2293906, *p* = 0.47 for rs4075943, *p* = 0.12 for rs13091808, *p* = 0.79 for rs178831, *p* = 0.95 for rs3759173, *p* = 0.10 for rs16865677, and *p* = 0.14 for rs3819122), while the *p*-values for rs12456284 and rs75704921 were 0.001 and 0.03, respectively.

For the case-control analysis, the whole case sample was subdivided into two subgroups: cases with two or more family members with BC and/or OC (n = 316) (subgroup A) and cases with sporadic early-onset BC (≤50 years) (n = 176) (subgroup B). Subgroup A excludes the subgroup B cases. In the single-locus analysis, no significant differences were observed in the genotype or allele distributions for rs75704921:C>T (*ARID2*), rs2229032:A>C (*ATR*), rs3735156:C>G (*KMT2C*), rs2276738:G>C (*MAP3K13*), rs2293906:C>T (*MAP3K13*), rs4075943:T>A (*MAP3K13*), rs13091808:C>T (*MAP3K13*), rs178831:G>A (*NCOR1*), or rs3759173:C>A (*TBX3*), either in the whole data set or in subgroups A or B (*p* > 0.05) (Appendix A).

rs832583:C>A is located in the driver gene *MAP3K1* [10,13,24]. The minor allele frequency (MAF) (allele A) was significantly lower in subgroup A (0.35) than in controls (0.40) (OR = 0.7 [95% CI 0.6–0.9] *p* = 0.01). This result indicates that the A allele is associated with a protective effect against BC risk. We also observed a protective effect for A-allele carriers (C/A+A/A) in subgroup A (OR = 0.7 [95% CI 0.5–0.9] *p* = 0.01) (Table 2). No association was found between rs832583 and non-familial early-onset BC (≤50 years). We also assessed for a protective effect of rs832583 according to the number of BC cases per family (Table 3). No protective effect was found in families with two BC/OC cases, nor in those with a strong family history of BC (Table 3). Therefore, the SNP rs832583-A seems to decrease risk in familial BC.

No significant association with BC risk was found for the SNP rs16865677:G>T, located in the *SF3B1* driver gene [17,28], in *BRCA1/2*-negative familial BC. However, in sporadic early-onset BC (≤50 years), the MAF (allele T) was significantly higher in cases (0.24) than controls (0.18) (OR = 1.4 [95% CI 1.0–1.8] *p* = 0.01) (Table 2). We also observed increased BC risk for T-allele carriers (G/T+T/T) among subgroup B cases (OR = 1.4 [95% CI 1.0–2.0] *p* = 0.01) (Table 2). This result indicates that the SNP rs16865677:G>T is a risk factor in non-familial early-onset BC (≤50 years) and that the MAF (allele T) is associated with increased BC risk in this group of patients.

In the *SMAD4* driver gene [17,28], we studied the association of two SNPs (rs3819122 and rs12456284) with familial BC risk. For rs3819122:A>C, the MAF (allele C) was higher in the whole sample (0.44) and in subgroup A (0.45) vs. controls (0.37) (OR = 1.3 [95% CI 1.1–1.5] *p* = 0.0005 and OR = 1.3 [95% CI 1.1–1.6] *p* = 0.0005, respectively). Furthermore, in the whole sample and in subgroup A, homozygous C/C individuals had significantly increased BC risk (OR = 1.5 [95% CI 1.1–2.1] *p* = 0.01 and OR = 1.7 [95% CI 1.2–2.5] *p* = 0.002, respectively). C-allele carriers (A/C+C/C) showed increased BC risk in the whole sample and in subgroup A (OR = 1.6 [95% CI 1.3–2.0] *p* ≤ 0.0001 and OR = 1.6 [95% CI 1.2–2.1] *p* = 0.0001, respectively) (Table 2), indicating that allele C is associated with elevated risk. When we analyzed the effect of the C allele by the number of BC cases per family, we observed a significantly increased BC risk for C/C homozygous individuals and for C-allele carriers (A/C+C/C) in families with two BC and/or OC cases (OR = 2.1 [95% CI 1.3–3.5] *p* = 0.001 and OR = 2.0 [95% CI 1.3–2.9] *p* ≤ 0.0001, respectively) (Table 3). Additionally, in sporadic cases diagnosed at ≤50 years of age (subgroup B), C-allele carriers (A/C+C/C) presented an increased BC risk (OR = 1.6 [95% CI 1.1–2.2] *p* = 0.006) (Table 2). These results allow us to conclude that SNP rs3819122:A>C elevates BC risk in familial *BRCA1/2*-negative BC in individuals with a moderate family history of BC and in non-familial early-onset BC (≤50 years).

With respect to rs12456284:A>G, the MAF frequency was higher in the whole sample (0.34) vs. controls (0.30) (OR = 1.2 [95% CI 1.0–1.3] *p* = 0.03) (Table 2). Moreover, we observed increased BC risk for G-allele carriers (A/G + G/G) in the whole sample, in subgroup A, and in subgroup B (OR = 1.3 [95% CI 1.0–1.6] *p* = 0.008; OR = 1.2 [95% CI 1.0–1.6] *p* = 0.04; OR = 1.4 [95% CI 1.0–1.9] *p* = 0.03, respectively). No significant effects were found according to the number of BC cases per family (Table 3). These results indicate that the G allele is associated with increased risk in familial BC and in single cases with early diagnosis (≤50 years).

Given that *SMAD4* rs3819122:A>C and *SF3B1* rs16865677:G>T were significantly associated with an increased risk of familial and non-familial early-onset BC (≤50 years), we evaluated the effect of composite genotype on risk. Table 4 shows the composite genotype frequencies for *SMAD4* rs3819122:A>C and *SF3B1* rs16865677:G>T. The A/C+G/G composite genotype showed a higher frequency in cases than controls in the whole sample and in subgroup A (OR = 1.3 [CI 95% 1.0–1.7] *p* = 0.045 and OR = 1.4 [CI 95% 1.0–2.0] *p* = 0.023, respectively). Another composite genotype associated with BC risk was A/C-G/T, which showed a higher frequency in cases compared to controls in the whole sample and in subgroup B (OR = 1.7 [CI 95% 1.2–2.4] *p* = 0.0013 and OR = 2.1 [CI 95% 1.3–3.5] *p* = 0.0021, respectively). The C/C-G/T composite genotype was associated with increased BC risk in the whole sample and in subgroup A (OR = 1.8 [CI 95% 1.1–3.0] *p* = 0.023 and OR = 2.2 [CI 95% 1.2–3.9] *p* = 0.012, respectively). When we analyzed the relationship between composite genotype and BC risk according to the number of BC cases per family (Table 5), we observed that the A/C-G/G, A/C-G/T, and C/C-G/T composite genotypes were associated with a significantly increased BC risk in cases belonging to families with moderate BC risk (OR = 1.7 [CI 95% 1.0–2.7] *p* = 0.019; OR = 1.8 [CI 95% 1.0–3.1] *p* = 0.037; OR = 2.4 [CI 95% 1.1–5.2] *p* = 0.03, respectively).

## 3. Discussion

The majority of both sporadic and familial BC cases are attributable to the accumulation of genetic mutations in the breast tissue throughout life. Some of these mutations, known as driver mutations, contribute to tumor progression [29]. Up to 90% of breast tumors may be caused by somatic driver mutations that initiate carcinogenic processes [12,30]. Most known driver genes were originally identified in sporadic breast tumors using NGS, including *ARID1B*, *CASP8*, *MAP3K1*, *NCOR1*, *SMARCD1*, *SMAD4*, *TBX3*, *SF3B1*, and *TBX3*. Recently, it has been proposed that inherited variants in driver genes could influence BC risk, progression, and/or survival. SNPs are the most common form of variation present in the human genome. In the present study, we evaluated the impact of 13 SNPs located in known or potential driver genes on familial and non-familial early-onset BC, in Chilean cases negative for *BRCA1/2* point mutations.

No significant differences between cases and controls were observed in terms of genotype or allele distributions for rs75704921:C>T (*ARID2*), rs2229032:A>C (*ATR*), rs3735156:C>G (*KMT2C*), rs2276738:G>C (*MAP3K13*), rs2293906:C>T (*MAP3K13*), rs4075943:T>A (*MAP3K13*), rs13091808:C>T (*MAP3K13*), rs178831:G>A (*NCOR1*), or rs3759173:C>A (*TBX3*), either in the whole data set or in subgroups A or B (*p* > 0.05). However, we found associations between the SNPs rs832583 (*MAP3K1*), rs16865677 (*SF3B1*), rs3819122 (*SMAD4*), and rs12456284 (*SMAD4*) and BC risk.

*MAP3K1* is a tumor suppressor [31,32] and cancer driver gene [33]. *MAP3K1* encodes a serine-threonine kinase in the *MAP3K* family which acts within the MAP-signaling pathway, triggering expression of genes important for angiogenesis, proliferation, and cell migration. *MAP3K1* is frequently mutated in human cancers, and mutations of this gene are the second-most prevalent genetic variation in BC [31]. Somatic mutations of *MAP3K1* are observed in 6% of BC cases, predominantly in ER-positive BC [34]. It has been suggested that MAP3K1 might act as an essential factor for promoting cell proliferation, migration, invasion, and drug resistance, increasing local recurrences and metastases of hormone receptor (HR)-positive/HER2-negative early-stage BC [34]. MAP3K1 overexpression plays a major role in poor prognosis in HR-positive, HER2-negative BC [34]. Importantly, SNPs appear to influence not only overall BC risk, but also the type of BC that develops in an individual. Unfortunately, association studies on *MAP3K1* SNPs and BC risk are scarce. Kuo et al. (2023) [34] reported that rs889312 (*MAP3K1*) is closely associated with poor disease-free survival and overall survival in early-stage HR-positive BC. We studied the SNP in *MAP3K1* rs832583:C>A, located at c.3190 in the coding region. There are no prior association studies in the literature on rs832583:C>A and BC risk. Our results suggested a protective effect against BC risk associated with the rs832583-A allele in patients with two or more family members with BC and/or OC. Interestingly, Li et al. (2022) [35] reported that elevated MAP3K1 expression is linked to longer survival time in ER+/HER2− and ER+/HER2+ BC patients. Therefore, it could be that the protective effect of the SNP rs832583 A allele is a consequence of increased MAP3K1 activity.

*SF3B1* is a driver gene that encodes subunit 1 of splicing factor 3b, a central spliceosome component important for anchoring the spliceosome to the precursor mRNA [36]. SF3B1 mutations are found in solid tumors such as BC, pancreatic carcinoma, uveal melanoma, and endometrial cancers [37]. SF3B1 is overexpressed in BC samples and is associated with lymph node metastasis. The most common SF3B1 mutation in BC is the K700E variant, followed by other recurrent mutations such as K666Q and K666E. All of these mutations are located in the coding region of the *SF3B1* gene [38]. It must be noted that BC is a complex set of pathologies rather than a single disease. The number and diversity of genomic drivers may explain the clinical heterogeneity of cases, and SNPs represent one such type of variation that can influence overall risk and type of BC. To date, there have been no association studies available on *SF3B1* SNPs and BC risk. The SNP rs16865677 is located within 2 kb of DNA immediately 5′ of the *SF3B1* coding sequence, a region that includes transcription start sites and major regulatory elements. Therefore, it is important to determine whether the SNP rs16865677 might contribute to BC risk. Our data indicated no association between the SNP rs16865677:G>T and BC risk in *BRCA1/2*-negative familial BC. Nevertheless, this SNP was shown to be a likely risk factor in non-familial early-onset BC (≤50 years), and the MAF (allele T) was associated with increased BC risk in this group of patients. There is little information in the literature regarding the genetic etiology of early-onset BC; therefore, the results obtained here are noteworthy. Considering the location of rs16865677:G>T, this SNP could enhance *SF3B1* expression, thereby increasing BC risk. Moreover, our results are consistent with Fu et al. [36], who reported that *SF3B1* mutations are associated with age at diagnosis, ER status, and histological grade in PR-negative patients, while in their luminal B subgroup, *SF3B1* mutations are associated only with age at diagnosis.

The *SMAD4* tumor suppressor gene encodes a protein involved in the transmission of chemical signals in the transforming growth factor-β (TGF-β) signal transduction pathway [39]. Abrogation of *SMAD4* function may result in the breakdown of the TGF-β-signaling pathway and loss of transcription of genes critical to cell cycle control [40]; as a result, cells may evade TGF-β-mediated growth control and apoptosis [41]. Recent studies have indicated that inactivation of *SMAD4* is related to disease progression in various cancers [42,43,44,45]. In colon cancer, *SMAD4* inactivation promotes malignancy and drug resistance [41]. In pancreatic cancer, low *SMAD4* expression is associated with malignant progression [40]. Similarly, in non-small-cell lung carcinoma, *SMAD4* expression has been shown to be higher in normal broncho-tracheal epithelium while lower in tumor tissues and closely correlated with lymph node metastasis. Thus, low *SMAD4* expression is linked to unfavorable outcomes in several types of cancers. With respect to BC, Lui et al. [39] reported that *SMAD4* expression appears to be decreased in BC cells vs. normal tissue, and reduced *SMAD4* expression tends to be associated with more poorly-differentiated tumors, a higher risk of recurrence, and shorter overall survival (OS). Kruijt et al. [46] also reported that low *SMAD4* expression is correlated with unfavorable prognosis for progression-free survival in BC patients. Woo et al. (2019) showed that high *SMAD4* expression in BC is positively associated with early stages, estrogen receptor positivity, and human epidermal growth factor receptor 2 negativity [47]. These authors also reported that a significant difference in OS is associated with high *SMAD4* expression in patients with T1 stage tumors. Liu et al. (2015) demonstrated a decreasing trend in *SMAD4* protein levels from histological grade I to III, suggesting that *SMAD4* may participate in the carcinogenesis and progression of breast ductal carcinoma. Patients with *SMAD4*-negative cancers exhibit a two-fold greater risk of recurrence and three-fold higher risk of mortality than those with *SMAD4*-positive cancers [39]. These results are consistent with Stuelten et al. [48] and Ren et al. [49], which also demonstrated that *SMAD4* protein expression is markedly downregulated or lost in breast ductal carcinoma when compared to the normal breast epithelium.

Despite these intriguing studies related to *SMAD4* protein expression, few publications have investigated the potential association between variants in driver or potential driver genes involved in the process of mammary tumorigenesis. In the *SMAD4* gene, we studied the association of two SNPs (rs3819122 and rs12456284) with familial BC risk. With respect to rs3819122, Marouf et al. (2016) found a strong association between rs3819122 (*SMAD4*) with BC tumor size (OR = 0.45, 95% CI 0.25–0.82, *p* = 0.009) [46]. Nevertheless, there are no studies in the literature that associate this SNP with BC risk. Our results for rs3819122:A>C (*SMAD4*) showed that this SNP increased BC risk in familial *BRCA1/2*-negative BC and in non-familial early-onset BC (≤50 years) [50]. Considering that rs3819122 is located in the 3′UTR of the *SMAD4* gene [50], this SNP could decrease the stability of *SMAD4* mRNA, causing pre-mRNA degradation and decreased *SMAD4* protein levels. With respect to rs12456284:A>G, Marouf et al. (2017) observed an increased risk of BC (OR = 2.04, 95% CI 1.32–3.15, *p* = 0.001) in a Moroccan population, which is consistent with our results showing that this SNP was associated with increased BC risk in familial BC and in single cases with early diagnosis (≤50 years) [50]. The SNP rs12456284 is also located in the 3′UTR of the *SMAD4* gene and has been predicted to influence potential miRNA binding and downregulate *SMAD4* gene expression [51,52]. It is important to highlight the association of the SNPs rs3819122 and rs12456284 with increased risk in single BC cases with early diagnosis (≤50 years), a disease for which there is scarce information on genetic variation and risk. However, functional studies are necessary to verify the influence of these polymorphisms on BC risk.

Given that *SMAD4* rs3819122:A>C and *SF3B1* rs16865677:G>T were significantly associated with an increased BC risk in familial/non-familial early-onset BC (≤50 years), we evaluated the effect of composite genotype on BC risk. The composite genotype analysis suggests that these two variants could produce an additive effect on BC risk.

There are scarce publications in the literature that provide analyses similar to those performed in the present manuscript.

## 4. Materials and Methods

### 4.1. Families

A total of 492 *BRCA1/2*-negative BC patients from Chilean families were enrolled from Corporación Nacional del Cancer (CONAC) files. None of the families fulfilled criteria for other BC-related syndromes, including ataxia-telangiectasia, Li-Fraumeni, or Cowden disease. All index cases were tested for *BRCA1/2* point mutations using Axen BRCA Macrogen. Pedigrees were constructed on the basis of an index case considered to have the highest probability of being a deleterious mutation carrier.

All families participating in the study were of self-reported Chilean ancestry dating from several generations, confirmed by extensive interviews with several members of each family from different generations. The study was approved by the Institutional Review Board of the University of Chile, School of Medicine (Project code number 1200049, March 2020). Informed consent was obtained from all participants.

### 4.2. Control Population

The control group of healthy Chilean individuals (n = 1285) was recruited from CONAC files. DNA samples were taken from unrelated individuals with no personal or family history of cancer who consented to anonymous testing. These individuals were interviewed and informed as to the aims of the study. DNA samples were obtained in accordance with all ethical and legal requirements. The control sample was matched by age and socioeconomic strata with respect to the cases.

### 4.3. SNPs Selection

We selected potential BC driver genes based on the methodology described by Göhler et al. (2017) [24]. Briefly, this methodology is focused on genes described to carry BC driver mutations in at least two of the following publications: Banerji et al. (2012) [21]; Ellis et al. (2012) [22]; Shah et al. (2012) [23]; Stephens et al. (2012) [14]. Well-known and intensively-studied genes such as *BRCA1*, *BRCA2*, *TP53*, and *PTEN* were excluded from the selection. SNP selection was performed using the Ensembl Genome browser and was based on the following criteria: (1) minor allele frequency (MAF) value over 10%; (2) location within the coding region (non-synonymous SNPs), core promoter region, or 5′- and 3′-untranslated regions (UTRs); (3) linkage disequilibrium (LD; r2 ≥ 0.80), determined using Haploview, to minimize the number of SNPs to be genotyped.

### 4.4. Genotyping Analysis

Genomic DNA was extracted from peripheral blood lymphocytes of the 492 cases belonging to the selected high-risk families and the 1285 controls. Samples were obtained according to the methods described by Chomczynski and Sacchi [53].

Genotyping of the SNPs rs832583 (*MAP3K1*), rs16865677 (*SF3B1*), rs3819122 (*SMAD4*), rs12456284 (*SMAD4*), rs75704921 (*ARID2*), rs2229032 (*ATR*), rs3735156 (*KMT2C*), rs2276738 (*MAP3K13*), rs2293206 (*MAP3K13*), rs4075943 (*MAP3K13*), rs13091808 (*MAP3K13*), rs178831 (*NCOR1*), and rs3759173 (*TBX3*) was performed using the commercially-available TaqMan Genotyping Assay (Applied Biosystems, Foster City, CA, USA) (assay IDs C___8961436_1_, C__33292283_10, C__27494402_10, C___2688971_10, C_104537382_10, C__26021082_10, C_____32482_1_, C__15882373_10, C___1832912_20, C__26073928_10, C___1716977_10, C____597484_10, and C__31115971_10, respectively). The reaction was performed in a 10 µL final volume containing 5 ng of genomic DNA, 1X TaqMan Genotyping Master Mix, and 20X TaqMan SNP Genotyping Assay. The polymerase chain reaction was carried out in a StepOnePlus Real-Time PCR System (Applied Biosystems, Foster City, CA, USA). The thermal cycles were initiated for 10 min at 95 °C, followed by 40 cycles each of 92 °C for 15 s and 60 °C for 1 min. Each genotyping run contained control DNA confirmed by sequencing. The alleles were assigned using StepOne software v 2.2 (Applied Biosystems, Foster City, CA, USA). As a quality control, we repeated the genotyping on ~10% of the samples, and all genotype scoring was performed and checked separately by two reviewers unaware of case-control status.

### 4.5. Statistical Analysis

The Hardy–Weinberg equilibrium assumption was assessed in the control sample using a goodness-of-fit chi-square test (HW Chisq function included in the “Hardy Weinberg” package v 1.4.1 for R, Foundation for Statistical Computing, Vienna, Austria, https://www.r-project.org/ accessed on 12 February 2023). Fisher’s exact test was used to test the association between the genotypes/alleles and case-control status. Odds ratios (ORs) with 95% confidence intervals (CIs) were calculated to estimate the strength of the associations. Odds ratio and Fisher’s exact test functions were performed using GraphPad Prism v 6.0 for Windows 10 (GraphPad Software, La Jolla, CA, USA, https://www.graphpad.com). A two-tailed *p*-value < 0.05 was used as the criterion of significance.

## Figures and Tables

**Table 1 ijms-24-16076-t001:** Inclusion criteria for the enrolled families.

Inclusion Criteria	Families: n (%)
Three or more family members with breast and/or ovarian cancer	148 (29.8%)
Two family members with breast and/or ovarian cancer	166 (33.6%)
Single affected individual with breast cancer age ≤35	87 (17.9%)
Single affected individual with breast cancer age 36–50	91 (18.7%)
Total	492 (100%)

**Table 2 ijms-24-16076-t002:** Genotype and allele frequencies of rs832583 (*MAP3K1)*, rs16865677 (*SF3B1*), rs3819122 (*SMAD4*), and rs12456284 (*SMAD4*) in *BRCA1/2*-negative breast cancer cases and controls.

Genotype or Allele	Controls (%) (n = 1285)	All BC Cases (n = 492)			Families with ≥2 BC and/or OC Cases (n = 316)			Families with a Single Case, Diagnosis at ≤50 Years of Age (n = 176)		
		BC Cases (%)	OR [95% CI]	*p*-Value ^a^	BC Cases (%)	OR [95% CI]	*p*-Value ^a^	BC Cases (%)	OR [95% CI]	*p*-Value ^a^
rs832583 (*MAP3K1)*										
C/C	473 (36.8)	198 (40.3)	1.0 (ref)	-	140 (44.3)	1.0 (ref)	-	59 (33.3)	1.0 (ref)	-
C/A	587 (45.7)	219 (44.5)	0.8 [0.6–1.0]	0.32	**132 (41.7)**	**0.7 [0.5–0.9]**	**0.04**	86 (49.2)	1.1 [0.8–1.5]	0.42
A/A	225 (17.4)	75 (15.2)	0.7 [0.5–1.0]	0.14	**44 (14.0)**	**0.6 [0.4–0.9]**	**0.03**	31 (17.5)	1.1 [0.7–1.7]	0.72
C/A+A/A	812 (63.2)	294 (59.7)	0.8 [0.6–1.0]	0.18	**176 (55.7)**	**0.7 [0.5–0.9]**	**0.01**	117 (66.7)	1.1 [0.8–1.6]	0.40
Allele C	1533 (59.6)	615 (62.5)	1.0 (ref)	-	412 (65.2)	1.0 (ref)	-	204 (57.9)	1.0 (ref)	-
Allele A	1037 (40.4)	369 (37.5)	0.8 [0.7–1.0]	0.14	**220 (34.8)**	**0.7 [0.6–0.9]**	**0.01**	148 (42.1)	1.0 [0.8–1.3]	0.56
rs16865677 (*SF3B1*)										
G/G	870 (67.7)	319 (64.9)	1.0 (ref)	-	216 (68.5)	1.0 (ref)	-	103 (58.5)	1.0 (ref)	-
G/T	364 (28.3)	151 (30.6)	1.1 [0.8–1.4]	0.31	89 (27.9)	0.9 [0.7–1.2]	0.88	**62 (35.2)**	**1.4 [1.0–2.0]**	**0.03**
T/T	51 (4.0)	22 (4.5)	1.1 [0.7–2.0]	0.49	11 (3.6)	0.8 [0.4–1.6]	0.86	11 (6.3)	1.8 [0.9–3.5]	0.09
G/T+T/T	415 (32.3)	173 (35.1)	1.1 [0.9–1.4]	0.25	100 (31.5)	0.9 [0.7–1.2]	0.83	**73 (41.5)**	**1.4 [1.0–2.0]**	**0.01**
Allele G	2104 (81.9)	789 (80.1)	1.0 (ref)	-	521 (82.5)	1.0 (ref)	-	268 (76.1)	1.0 (ref)	-
Allele T	466 (18.1)	195 (19.9)	1.1 [0.9–1.4]	0.11	111 (17.5)	0.9 [0.7–1.2]	0.76	**84 (23.9)**	**1.4 [1.0–1.8]**	**0.01**
rs3819122 (*SMAD4*)										
A/A	519 (40.4)	143 (29.1)	1.0 (ref)	-	91 (28.8)	1.0 (ref)	-	52 (29.8)	1.0 (ref)	-
A/C	576 (44.8)	**269 (54.5)**	**1.6 [1.3–2.1]**	**<0.0001**	**166 (52.4)**	**1.6 [1.2–2.1]**	**0.0005**	**103 (58.4)**	**1.7 [1.2–2.5]**	**0.001**
C/C	190 (14.8)	**80 (16.3)**	**1.5 [1.1–2.1]**	**0.01**	**59 (18.8)**	**1.7 [1.2–2.5]**	**0.002**	21 (11.8)	1.1 [0.6–1.8]	0.77
A/C+C/C	766 (59.6)	**349 (70.9)**	**1.6 [1.3–2.0]**	**<0.0001**	**225 (71.2)**	**1.6 [1.2–2.1]**	**0.0001**	**124 (70.2)**	**1.6 [1.1–2.2]**	**0.006**
Allele A	1614 (62.8)	555 (56.4)	1.0 (ref)	-	348 (55.0)	1.0 (ref)	-	207 (59.0)	1.0 (ref)	-
Allele C	956 (37.2)	**429 (43.6)**	**1.3 [1.1–1.5]**	**0.0005**	**284 (45.0)**	**1.3 [1.1–1.6]**	**0.0005**	145 (41.0)	1.1 [0.9–1.4]	0.20
rs12456284 (*SMAD4*)										
A/A	685 (53.3)	227 (46.2)	1.0 (ref)	-	148 (46.9)	1.0 (ref)	-	79 (45.1)	1.0 (ref)	-
A/G	427 (33.2)	**196 (39.9)**	**1.3 [1.0–1.7]**	**0.005**	**122 (38.7)**	**1.3 [1.0–1.7]**	**0.04**	**74 (41.8)**	**1.5 [1.0–2.1]**	**0.02**
G/G	173 (13.5)	69 (13.9)	1.2 [0.8–1.6]	0.24	46 (14.4)	1.2 [0.8–1.7]	0.28	23 (13.1)	1.1 [0.7–1.8]	0.60
A/G+G/G	600 (46.7)	**265 (53.8)**	**1.3 [1.0–1.6]**	**0.008**	**168 (53.1)**	**1.2 [1.0–1.6]**	**0.04**	**97 (54.9)**	**1.4 [1.0–1.9]**	**0.03**
Allele A	1797 (69.9)	650 (66.1)	1.0 (ref)	-	418 (66.1)	1.0 (ref)	-	232 (66.0)	1.0 (ref)	-
Allele G	773 (30.1)	**334 (33.9)**	**1.2 [1.0–1.3]**	**0.03**	214 (33.9)	1.1 [0.9–1.4]	0.07	120 (34.0)	1.2 [0.9–1.5]	0.14

BC: breast cancer, OC: ovarian cancer, OR: odds ratio, CI: confidence interval; ^a^ Fisher’s exact test; bold values are statistically significant (*p* < 0.05).

**Table 3 ijms-24-16076-t003:** Genotype and allele frequencies of rs832583 (*MAP3K1)*, rs16865677 (*SF3B1*), rs3819122 (*SMAD4*) and rs12456284 (*SMAD4*) according to the number of BC cases per family in *BRCA1/2*-negative breast cancer cases and controls.

Genotype or Allele	Controls (%) (n = 1285)	Families with ≥2 BC and/or OC Cases (n = 165)			Families with ≥3 BC and/or OC Cases (n = 151)		
		BC Cases (%)	OR [95% CI]	*p*-Value ^a^	BC Cases (%)	OR [95% CI]	*p*-Value ^a^
rs832583 (*MAP3K1)*							
C/C	473 (36.8)	73 (44.2)	1.0 (ref)	-	67 (44.3)	1.0 (ref)	-
C/A	587 (45.7)	69 (41.7)	0.7 [0.5–1.0]	0.14	63 (41.6)	0.7 [0.5–1.0]	0.13
A/A	225 (17.4)	23 (14.1)	0.7 [0.4–1.1]	0.12	21 (14.1)	0.6 [0.3–1.1]	0.11
C/A+A/A	812 (63.2)	92 (55.8)	0.7 [0.5–1.0]	0.07	84 (55.7)	0.7 [0.5–1.0]	0.07
Allele C	1533 (59.6)	215 (65.0)	1.0 (ref)	-	197 (65.1)	1.0 (ref)	-
Allele A	1037 (40.4)	115 (35.0)	0.7 [0.6–1.0]	0.06	105 (34.9)	0.7 [0.6–1.0]	0.06
rs16865677 (*SF3B1*)							
G/G	870 (67.7)	114 (69.0)	1.0 (ref)	-	103 (68.0)	1.0 (ref)	-
G/T	364 (28.3)	45 (27.3)	0.9 [0.6–1.3]	0.85	43 (28.5)	1.0 [0.6–1.4]	1.0
T/T	51 (4.0)	6 (3.7)	0.9 [0.4–2.0]	1.0	5 (3.5)	0.8 [0.3–2.1]	1.0
G/T+T/T	415 (32.3)	51 (31.1)	0.9 [0.6–1.3]	0.78	48 (31.9)	0.9 [0.6–1.4]	1.0
Allele G	2104 (81.9)	273 (82.6)	1.0 (ref)	-	249 (82.3)	1.0 (ref)	-
Allele T	466 (18.1)	57 (17.4)	0.9 [0.7–1.2]	0.81	53 (17.7)	0.9 [0.7–1.3]	0.93
rs3819122 (*SMAD4*)							
A/A	519 (40.4)	41 (25.0)	1.0 (ref)	-	50 (32.9)	(ref)	-
A/C	576 (44.8)	**91 (55.3)**	**2.0 [1.3–2.9]**	**0.0004**	74 (49.3)	1.3 [0.9–2.0]	0.13
C/C	190 (14.8)	**33 (19.7)**	**2.1 [1.3–3.5]**	**0.001**	27 (17.9)	1.4 [0.8–2.4]	0.16
A/C+C/C	766 (59.6)	**124 (75.0)**	**2.0 [1.3–2.9]**	**<0.0001**	101 (67.1)	1.3 [0.9–2.0]	0.09
Allele A	1614 (62.8)	173 (52.6)	1.0 (ref)	-	174 (57.5)	(ref)	-
Allele C	956 (37.2)	**157 (47.4)**	**1.5 [1.2–1.9]**	**0.003**	128 (42.5)	1.2 [0.9–1.6]	0.08
rs12456284 (*SMAD4*)							
A/A	685 (53.3)	77 (46.9)	1.0 (ref)	-	71 (46.8)	1.0 (ref)	-
A/G	427 (33.2)	64 (38.6)	1.3 [0.9–1.8]	0.11	59 (38.9)	1.3 [0.9–1.9]	0.12
G/G	173 (13.5)	24 (14.5)	1.2 [0.7–2.0]	0.48	21 (14.3)	1.1 [0.6–1.9]	0.58
A/G+G/G	600 (46.7)	88 (53.1)	1.3 [0.9–1.8]	0.11	80 (53.2)	1.2 [0.9–1.8]	0.14
Allele A	1797 (69.9)	192 (66.2)	1.0 (ref)	-	201 (66.6)	1.0 (ref)	-
Allele G	773 (30.1)	98 (33.8)	1.1 [0.9–1.5]	0.21	101 (33.4)	1.1 [0.9–1.5]	0.25

BC: breast cancer, OC: ovarian cancer, OR: odds ratio, CI: confidence interval; ^a^ Fisher’s exact test; bold values are statistically significant (*p* < 0.05).

**Table 4 ijms-24-16076-t004:** Composite genotype frequencies for rs3819122 (*SMAD4*) and rs16865677 (*SF3B1*) in *BRCA1/2*-negative breast cancer cases and controls.

Composite Genotypes			All BC Cases (n = 488)			Families with ≥2 BC and/or OC Cases (n = 314)			Families with a Single Case, Diagnosis at ≤50 Years of Age (n = 174)		
*SMAD4*	*SF3B1*	Controls n = 1261 (%)	BC Cases (%)	OR [95% CI]	*p*-Value ^a^	BC Cases (%)	OR [95% CI]	*p*-Value ^a^	BC Cases (%)	OR [95% CI]	*p*-Value ^a^
rs3819122	rs16865677										
A/A	G/G	336 (26.62)	106 (21.72)	1.0 (ref)	-	66 (21.02)	1.0 (ref)	-	40 (22.99)	1.0 (ref)	-
A/A	G/T	153 (12.12)	34 (6.97)	0.7 [0.4–1.1]	0.12	20 (6.37)	0.6 [0.3–1.1]	0.16	14 (8.05)	0.7 [0.4–1.4]	0.53
A/A	T/T	22 (1.74)	9 (1.84)	1.3 [0.5–2.9]	0.52	5 (1.59)	0.4 [0.1–2.0]	0.39	4 (2.30)	1.5 [0.5–4.6]	0.51
A/C	G/G	**390 (30.90)**	**165 (33.81)**	**1.3 [1.0–1.7]**	**0.045**	**113 (35.99)**	**1.4 [1.0–2.0]**	**0.023**	52 (29.89)	1.1 [0.7–1.7]	0.65
A/C	G/T	**155 (12.28)**	**86 (17.62)**	**1.7 [1.2–2.4]**	**0.0013**	46 (14.65)	1.5 [0.9–2.3]	0.059	**40 (22.99)**	**2.1 [1.3–3.5]**	**0.0021**
A/C	T/T	23 (1.82)	12 (2.46)	1.6 [0.7–3.4]	0.22	6 (1.91)	1.3 [0.5–3.3]	0.60	6 (3.45)	2.1 [0.8–5.7]	0.12
C/C	G/G	130 (10.30)	47 (9.63)	1.1 [0.7–1.7]	0.53	37 (11.78)	1.4 [0.9–2.2]	0.12	10 (5.75)	0.6 [0.3–1.3]	0.31
C/C	G/T	**48 (3.80)**	**28 (5.74)**	**1.8 [1.1–3.0]**	**0.023**	**21 (6.69)**	**2.2 [1.2–3.9]**	**0.012**	7 (4.02)	1.2 [0.5–2.8]	0.64
C/C	T/T	5 (0.40)	1 (0.20)	0.6 [0.07–5.49]	1.00	0 (0.00)	0.46 [0.02–8.42]	1.00	1 (0.57)	1.6 [0.1–14.8]	0.49

BC: breast cancer, OC: ovarian cancer, OR: odds ratio, CI: confidence interval; ^a^ Fisher’s exact test; bold values are statistically significant (*p* < 0.05).

**Table 5 ijms-24-16076-t005:** Composite genotype frequencies for rs3819122 (*SMAD4*) and rs16865677 (*SF3B1*) according to the number of BC cases per family in *BRCA1/2*-negative breast cancer cases and controls.

Composite Genotypes			Families with ≥2 BC and/or OC Cases (n = 163)			Families with ≥3 BC and/or OC Cases (n = 151)		
*SMAD4*	*SF3B1*	Controls n = 1261 (%)	BC Cases (%)	OR [95% CI]	*p*-Value ^a^	BC Cases (%)	OR [95% CI]	*p*-Value ^a^
rs3819122	rs16865677							
A/A	G/G	336 (26.62)	31 (19.02)	1.0 (ref)	-	35 (23.18)	1.0 (ref)	-
A/A	G/T	153 (12.12)	7 (4.29)	0.4 [0.2–1.1]	0.10	13 (8.61)	0.8 [0.4–1.5]	0.63
A/A	T/T	22 (1.74)	3 (1.84)	1.4 [0.4–5.2]	0.46	2 (1.32)	0.8 [0.1–3.8]	1.00
A/C	G/G	390 (30.90)	**62 (38.04)**	**1.7 [1.0–2.7]**	**0.019**	51 (33.77)	1.2 [0.7–1.9]	0.36
A/C	G/T	155 (12.28)	**26 (15.95)**	**1.8 [1.0–3.1]**	**0.037**	20 (13.25)	1.2 [0.6–2.2]	0.54
A/C	T/T	23 (1.82)	3 (1.84)	1.4 [0.4–4.9]	0.48	3 (1.99)	1.2 [0.3–4.3]	0.72
C/C	G/G	130 (10.30)	20 (12.27)	1.6 [0.9–3.0]	0.10	17 (11.26)	0.6 [0.3–1.3]	0.31
C/C	G/T	48 (3.80)	**11 (6.75)**	**2.4 [1.1–5.2]**	**0.03**	10 (6.62)	2.0 [0.9–4.3]	0.10
C/C	T/T	5 (0.40)	0 (0.00)	0.9 [0.05–18.00]	1.00	0 (0.00)	0.8 [0.04–15.90]	1.00

BC: breast cancer, OC: ovarian cancer, OR: odds ratio, CI: confidence interval; ^a^ Fisher’s exact test; bold values are statistically significant (*p* < 0.05).

## Data Availability

All data are shown within the manuscript.

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
