# Peer review of "Association of Germline Variation in Driver Genes with Breast Cancer Risk in Chilean Population"

_ijms, 2023, doi:10.3390/ijms242216076_

Round 1

Reviewer 1 Report

Comments and Suggestions for Authors

The study investigates the association between specific SNPs in known or potential driver genes and breast cancer risk, which is a relevant and important area of research. The focus on familial and non-familial early-onset breast cancer in the Chilean population can contribute to the understanding of genetic factors in breast cancer susceptibility. The manuscript conforms to the journal's specified section order for a research paper.

Abstract Section appears to be well-organised and offers a clear and succinct summary of your research. 

Introduction Section is well-structured and informative, but there are a few points the authors could consider adding to provide a more comprehensive background for your research:

- the authors mentioned the global impact of breast cancer briefly, but they could include some additional key statistics or trends related to breast cancer on a global scale. This could further emphasise the significance of their research.

- the authors have discussed the incidence and mortality rates of breast cancer in Chile, which is excellent. However, they might want to include a brief overview of the current breast cancer healthcare landscape in Chile, such as screening programs, treatment options, or any specific challenges that patients in Chile face.

- the authors have mentioned the major susceptibility genes like BRCA1 and BRCA2, but they could briefly touch upon what is currently known about them, especially in the Chilean context, to provide context for your study.

The "Materials and Methods" section appears to be clear and complete, outlining the procedures and methods used in the study. However, I would recommend a few minor improvements for clarity and consistency:

- in the "Control population" paragraph, consider specifying the range of ages and socioeconomic strata for the control group, as this information can be relevant for understanding the study's design.

Author Response

Please also find below our responses for the first reviewer. I would like to thank the reviewers for providing thorough notes on our manuscript and the journal for orchestrating a timely peer review.

1.- The authors briefly mention the global impact of breast cancer, but could include some additional key statistics or trends related to breast cancer on a global scale. This could further emphasize the importance of your research.

Answer: Thank you for this excellent recommendation. We added global data and predictions to the Introduction: “In women, BC accounts for approximately a quarter of all cancer cases and a sixth of all cancer deaths, ranking first for incidence in 159 of 185 countries and for mortality in 110 countries (Yizhen Li, et al Front Oncol. 2022; 12: 891824.). The incidence of female breast cancer is predicted to continue to increase globally, from 45.26 in 2010 to 47.36 in 2035 (Yizhen Li, Front Oncol. 2022; 12: 891824.)”.

2.- The authors have discussed the incidence and mortality rates of breast cancer in Chile, which is excellent. However, they might want to include a brief overview of the current breast cancer healthcare landscape in Chile, such as screening programs, treatment options, or any specific challenges that patients in Chile face.

Answer: Thank you very much for your suggestion. There are various screening programs, the most common being mammography. In fact, current laws provide for women to have annual mammograms without prior medical authorization in order to expedite the process. However, we believe that this information lies beyond the scope of the current manuscript because the focus is more genetic and biological than clinical. Therefore, we believe it is better to leave this context out of the manuscript.

3.- The authors have mentioned the main susceptibility genes such as BRCA1 and BRCA2, but could briefly address what is currently known about them, especially in the Chilean context, to provide context for their study.

Answer: We added data to the Introduction, with references regarding BRCA1/BRCA2 mutations in Chile:

Analysis of BRCA1/BRCA2 mutations currently plays a highly significant role in oncological clinical genetics worldwide, with the goal of improving prevention and treatment for women at high risk. In Chile, studies have been carried out in relation to prevalent mutations in these genes, and this information is used in clinical practice (Jara et al, 2017; Alvarez et al 2017 and 2022). Nevertheless, the major susceptibility genes only explain about 16% of risk.”

4.- In the "Control population" paragraph, consider specifying the range of ages and socioeconomic strata for the control group, as this information can be relevant for understanding the study's design.

Answer:  Thank you very much for your suggestion. We have the necessary information to build a table with the suggested data. However, all data must be collected from the informed consent paperwork, which unfortunately is not feasible given the tight deadline for submitting the review.

Reviewer 2 Report

Comments and Suggestions for Authors

In the presented manuscript the Authors examined the Association of germline variation in driver genes with breast cancer risk in Chilean population. It is an interesting topic. However, I have a few questions/comments.

1. The introduction was well-prepared, providing context and a clear purpose for the study.

2. I have no doubts about the methodology, results and conclusion presentation.

3. I think the Authors could describe the basis for selecting the SNPs in more detail. The Authors presented it very generally. Were any databases used for the criteria presented? What tools were used for this?

4. In my opinion, mutation data should not be related to polymorphism results (“Moreover, our results are consistent with Fu et al. [32], who reported that SF3B1 mutations are associated with age at diagnosis, ER status, and histological grade in PR-negative patients, while in their luminal B subgroup, SF3B1 mutations are associated only with age at diagnosis.”).

5. Is there a difference between p and p-value? I propose to standardize this throughout the text of the article

Author Response

Please find below our responses for the second reviewer. I would like to thank the reviewers for providing thorough notes on our manuscript and the journal for orchestrating a timely peer review. 

1.- I think the Authors could describe the basis for selecting the SNPs in more detail. The Authors presented it very generally. Were any databases used for the criteria presented? What tools were used for this?

Answer: Thank you very much for this valuable observation. We added the following paragraph to the Results section to explain the selection of SNPs:

“We selected potential BC driver genes based on the methodology described by Göhler et al. (2017). Briefly, this methodology is focused on genes described to carry BC driver mutations in at least two of the following publications: Banerji et al. (2012); Ellis et al. (2012); Shah et al. (2012); and Stephens et al. (2012). Well-known and intensively-studied genes such as BRCA1, BRCA2, TP53, and PTEN were excluded from selection. SNP selection was performed using the Ensembl Genome browser and was based on the following criteria: 1) minor allele frequency (MAF) value over 10%; 2) location within the coding region (non-synonymous SNPs), core promoter region, or 5’- and 3’- untranslated regions (UTRs); and 3) linkage disequilibrium (LD; r2 ≥ 0.80), determined using Haploview, to minimize the number of SNPs to be genotyped.”

2.- In my opinion, mutation data should not be related to polymorphism results (“Furthermore, our results are consistent with Fu et al. [32], who reported that mutations in SF3B1 are associated with age in the time of diagnosis, ER status and histological grade). in PR-negative patients, while in their luminal B subgroup, SF3B1 mutations are associated only with age at diagnosis”).

Answer: I do not agree with this perspective. SNPs are changes in the DNA and are therefore mutations if their frequency is less than 1%. The Discussion paragraph refers to the fact that both SNPs and mutations in the gene SF3B1 are associated with early-onset BC.

3.- Is there a difference between p and p-value? I propose to standardize this throughout the text of the article

Answer: Thank you for the observation. p and p-value are the same, which is commonly used as statistical nomenclature. To avoid confusion, we specify within the text that p corresponds to p-value.